# LiveCellMiner: A new tool to analyze mitotic progression

Daniel Moreno-Andrés[1☯]*, Anuk Bhattacharyya[2], Anja Scheufen[1], Johannes Stegmaier[2☯]*

**1** Institute of Biochemistry and Molecular Cell Biology, Medical School, RWTH Aachen University, Aachen, Germany, **2** Institute of Imaging and Computer Vision, RWTH Aachen University, Aachen, Germany

☯ These authors contributed equally to this work.
* dmoreno@ukaachen.de (DMA), johannes.stegmaier@lfb.rwth-aachen.de (JS)

**Data Availability Statement:** All described code is available at https://github.com/stegmaierj/LiveCellMiner and we created a snapshot of the state at submission at https://zenodo.org/badge/latestdoi/269630703. Example data to test all

## Abstract

Live-cell imaging has become state of the art to accurately identify the nature of mitotic and cell cycle defects. Low- and high-throughput microscopy setups have yield huge data amounts of cells recorded in different experimental and pathological conditions. Tailored semi-automated and automated image analysis approaches allow the analysis of high-content screening data sets, saving time and avoiding bias. However, they were mostly designed for very specific experimental setups, which restricts their flexibility and usability. The general need for dedicated experiment-specific user-annotated training sets and experiment-specific user-defined segmentation parameters remains a major bottleneck for fully automating the analysis process. In this work we present LiveCellMiner, a highly flexible open-source software tool to automatically extract, analyze and visualize both aggregated and time-resolved image features with potential biological relevance. The software tool allows analysis across high-content data sets obtained in different platforms, in a quantitative and unbiased manner. As proof of principle application, we analyze here the dynamic chromatin and tubulin cytoskeleton features in human cells passing through mitosis highlighting the versatile and flexible potential of this tool set.

## Introduction

State of the art cell cycle and mitosis research strongly relies on advanced live-cell microscopy for recording cells in model organisms and tissue cultures. With the help of different labeling techniques (target protein fusion with fluorescent proteins, direct fluorescent labeling of cellular targets, organelle specific fluorescent probes) subcellular structures and cell cycle markers can be followed by high-content screening (HCS) approaches. This allows investigating the different steps of life and fate of single cells and cell populations. Such experiments generate massive amounts of data, which help to pinpoint the nature of mitotic and cell cycle defects and to accurately identify and characterize key molecular factors in different experimental conditions and clinically relevant situations (for review, see [1–4]).

functions of the software are provided in this repository. All LiveCellMiner project files and spreadsheets that were used for generating the figures and tables in the paper are available from https://bitbucket.org/jstegmaier/livecellminer/downloads/LiveCellMiner_FileS3.zip. Original image data are on the order of multiple terabytes but can be made available on demand.

**Funding:** This work was funded by the German Research Foundation (Deutsche Forschungsgemeinschaft, DFG) under grant no. ME3737/3-1 and by the Excellence Initiative of the German federal and state governments in the form of an RWTH StartUp project (AB). The work with the Zeiss LSM710 confocal laser-scanning microscope was supported by the Confocal Microscopy Facility, a Core Facility of the Interdisciplinary Center for Clinical Research (IZKF) Aachen within the Faculty of Medicine at RWTH Aachen University.

**Competing interests:** The authors have declared that no competing interests exist.

Analyzing high-content data sets is a formidable task where supervised machine learning methods have been so far crucial [5–12]. More recently, convolutional neural networks (CNN) further improved the possibilities for automatic detection, segmentation and classification tasks [13–16]. Supervised machine learning approaches, however, are still time-consuming because they rely on user-curated phenotype definitions and analysis. To overcome the limitations, unsupervised machine learning without the need of manually-annotated morphology and phenotype-based classifications recently demonstrated first promising results [17–20].

Supervised and unsupervised phenotypic profiling of live-cell microscopy data is emerging as a powerful tool for clinicians, pharmaceutical industry and biology research in general. Multiparametric data analysis at single-cell level allows the integration of up to hundreds of quantitative features to describe and distinguish phenotypes going far beyond traditional approaches based on the analysis of one or a few features in single-cell images. Phenotypic classifiers are extraordinarily useful for analyzing complex populations. But because classes are categorical entities, they cannot accurately quantify continuous time-dependent gradual changes, as usually assumed in existing tools [10]. Moreover, in most of the cases where numerous features are measured and integrated to generate high-dimensional phenotypic profiles, it is unclear whether all of them are required for describing the phenotypic classes. However, the quantification of simple morphological and shape parameters, *e.g.*, of the cell nucleus, has potential diagnostic value [21, 22]. In this regard, most available tools are able to measure health-related morphology profiles of tuneable complexity in static images of fixed cell samples ([23]; for review, see [3]). Moreover, general-purpose visual analytics tools like [24] allow a versatile analysis of HCS-derived features but are unsuitable for modeling temporal object dependencies and lack dedicated single-cell synchronization. Therefore, tools for unbiased and comprehensive analysis of image features directly reflecting time-dependent live-cell shape and morphology are urgently needed both in basic and translational research [25].

Another common limitation in the available HCS tools is the lack of user-independent segmentation settings. The selection of a significant number of object detection parameters depends on user choices. This could ultimately introduce bias into phenotype classifiers because class-determining algorithms (whether supervised or unsupervised) learn from image features that can be modulated by the segmentation process. As first step, automated cell identification has recently been achieved using deep learning, where CNNs are particularly applicable for the instance segmentation task [16, 26, 27]. These tools perform well when being trained with sufficiently different modalities [16]. However, they tend to behave unpredictably when being applied to data that significantly deviates from images seen during the training phase, and importantly, lack of integrated tools to subsequently analyze quantitatively the features and/or phenotypic profiles of live-cell image data sets.

Here, we introduce LiveCellMiner, a new open-source fully-automated software tool for the quantitative analysis and comparison of 2D+t microscopy images of fluorescently-labeled cells. The software allows automatic segmentation and tracking and extracts quantitative features for all tracked objects. It enables automatic temporal synchronization of extracted tracks and offers comprehensive data visualization and selection possibilities. LiveCellMiner was primarily developed to analyze mitotic phenotypes in cells but can be easily extended to other scenarios. Since mitosis is characterized by a succession of distinctive chromatin morphologies, prophase, prometaphase, metaphase, early anaphase, late anaphase and telophase, it offers an excellent multi-level benchmark for the study of cytologic and temporal phenotypes. In early mitosis, the nuclear envelope breaks down and the chromatin condenses generating individualized and rod-shaped chromosomes, which are captured by the mitotic spindle and segregated to sister chromatin masses during anaphase (for review, see [28, 29]). During telophase and early G1, the chromatin masses decondense allowing the reassembly of functional nuclei

able of gene transcription and genome replication (for review, see [30–32]). Here, we reanalyze published and unpublished data sets of human tissue culture cells passing mitosis generated with different live-cell imaging platforms showing high reproducibility between the systems. We confirm previously described mitotic alterations after downregulation of the Lysine Specific Demethylase (LSD1), RecQ-like Helicase 4 (RecQL4) and the Protein Phosphate PP2A-complex but identify also phenotypes that so far escaped our attention. We also reanalyzed some available data [33] from open repositories to demonstrate the suitability of LiveCellMiner as a powerful tool to investigate high-throughput screening studies.

## Design and implementation

We developed LiveCellMiner, a new open-source software tool for studying the cytology of cell division under different experimental conditions. LiveCellMiner is an extension package for the general-purpose data mining MATLAB toolbox SciXMiner [34] and makes use of some existing tools for object detection and segmentation [16, 35, 36] (see S7 Table for a detailed list of dependencies). In the following sections, we present the individual modules of LiveCellMiner and show how they are used for data import, feature extraction, cell trajectory synchronization, data selection and visualization. The presented proof of principle applications are based on existing data sets from previous publications [37–39] where image data were acquired using 2D+t widefield and confocal microscopy in different platforms (see S1 Text for details on the experimental setup).

### Data import

LiveCellMiner expects time series of 2D images (2D+t) of cells with chromatin-labeled nuclei as the primary and required input. For two-channel experiments, LiveCellMiner can optionally extract additional intensity-related features based on the nucleus segmentation. Both channels need to have the same number of frames, the same image dimensions and need to be aligned properly. The first step comprises the detection, segmentation and tracking of all cell nuclei. We adapted the Laplacian-of-Gaussian-based object detection method implemented in XPIWIT [35, 36] to perform automatic detection of nuclei centroids using a set of predefined processing pipelines adjusted for various image resolutions. Detected centroids are tracked using the methods described in [40]. Briefly, tracking is performed in a time-reversed manner starting with the last frame and by sequentially linking objects to their predecessors using hierarchical clustering with Ward's linkage criterion [41]. The cluster cut-off can either be explicitly specified based on prior knowledge or it can be determined heuristically as half the average distance of each object to its eight spatially nearest neighbors (default setting). If two objects with a different tracking id end up in the same cluster, a cell division event is annotated. After tracking is performed, mitotic trajectories are extracted that fulfil user-defined constraints like a minimum number of successfully tracked frames before and after the cell division occurred (by default we set this parameters to 30 and 60 frames before and after the cell division, respectively, which corresponds to 90 minutes and 180 minutes with the 3 minutes sampling intervals used in all our experiments). All detected centroids are then used to initialize the automatic segmentation of the cells. We provide both a classical and a deep learning-based solution to the segmentation. The classical segmentation method crops a square region surrounding the current detection. First, the image is median filtered for noise reduction ($5 \times 5$ window size) and then binarized using the arithmetic mean of a threshold identified by Otsu's method [42] and the minimum intensity observed in the center part of the patch. The modified version of Otsu's threshold is used to avoid degenerate segmentations where a dim cell residing in the center of the patch could potentially be removed if it is surrounded by more bright

objects. As the cell of interest is located in the center of the image patch, we initialize a seeded watershed with two seeds, one for the center cell and another one for the background and neighboring cells. The seeded watershed is applied on an inverted Euclidean distance map with intensity minima located in the centers of the nuclei. As an alternative to the classical segmentation pipeline, we integrated an interface to the recently published Cellpose algorithm [16]. Cellpose is automatically started and parameterized to extract all segments in each of the images. We then use the centroids of successfully tracked objects and crop the results of the Cellpose segmentation using the same region size as for the classical approach. In each patch, we only keep the single central cell to constrain the feature extraction to this region. Although Cellpose generally provides highly accurate segmentation results for the majority of the cells, we found that directly using Cellpose to replace the LoG-based object detection yielded less reliable tracking results. In some cases, Cellpose failed to properly segment rarely occurring cell shapes observed in meta- and anaphase. Thus, we additionally included a fallback option for cases where segmentations provided by Cellpose were missing and in these cases occasionally switch back to the classical segmentation method for individual cells. A quantitative assessment of the segmentation and tracking quality is provided in S1 Table and a qualitative demonstration of the detection and segmentation accuracy as well as exemplary erroneous detections are depicted in S11 Fig.

## Extraction of quantitative features and project fusion

The segmented image patches are subsequently used for feature extraction. In addition to classical 2D features like area, centroid, major and minor axes, orientation, circularity and intensity statistics, we extracted a set of Haralick texture features from the gray level co-occurrence matrix [43] (see S2 and S3 Tables for an overview of all available features). We empirically set the number of gray levels to 64, removed the background to foreground transitions from the co-occurrence matrices and computed average values obtained for the neighbor relations $[0, 1], [1, 0], [1, 1], [−1, 1]$. Among others, the Haralick features comprise measures of texture entropy, correlation, contrast and variance (we refer to [43] for a complete definition and explanation of the individual features). Finally, we apply a GoogLeNet pretrained on ImageNet database on each image patch to obtain CNN-features that are used for automatic synchronization of the cell trajectories [44]. In addition to the raw image snippets of all available channels, the corresponding segmentations, GoogLeNet features and feature time series of all valid trajectories are stored in a SciXMiner-compatible format. Additional meta information like microscope, experiment ID, plate number and experimental conditions are saved as well and can later be used by the flexible and powerful data selection possibilities of SciXMiner. Individual projects obtained for different positions can be fused to a single SciXMiner project, to analyze even large projects in a single and consistent project file. After projects have been imported to the SciXMiner format, additional features can be derived from the time series and single features. In addition to all available feature transformations that are available by default in SciXMiner [34, 45], we incorporated dedicated features for the analysis of cell behavior.

Time series can be smoothed with a variable window size using any of the methods implemented in MATLAB's smooth function to remove small deviations in the extracted feature values. Moreover, absolute feature values can be normalized to predefined events of the cell cycle, such as the average interphase feature value or the feature value of the first late anaphase frame. As different microscopes or acquisition settings produce notably different absolute feature values, these normalization procedures are beneficial to make time series comparable among different experiments and to compute relative recovery time series of cell properties like fluorescence intensity after mitosis. The rate of change for selected features at a particular

time point (*e.g.*, to estimate the initial recovery rate of time series features immediately after cell division) can be approximated by a linear regression of the feature values in a small temporal window. The slope of these regression curves is stored as a single feature for each cell. Finally, to obtain a proxy for interphase recovery, we added a dedicated recovery feature that measures the absolute percentage deviation of one or more features to their respective interphase mean value with 100% indicating full recovery (average percentage deviation if multiple features are selected). While arbitrary features can be combined to a custom recovery feature, we used the features area, minor axis length, mean intensity, intensity standard deviation for all figures in this paper.

## Cell trajectory synchronization

To compensate for different duration of pro-, prometa- and metaphase, as observed also in our data sets, synchronization can be adapted to every particular cell cycle event under study. The LiveCellMiner toolbox provides different ways of synchronizing the individual trajectories. We chose two characteristic events as the synchronization anchors, to obtain properly aligned time series for quantitative comparisons. To this end, we identify interphase to prophase transition (IP) or metaphase/early anaphase to late anaphase transition (MA) as reference mark for alignment of interphase or postmitotic frames, respectively. There are currently three options for automatic alignment. The first approach uses the classical object features area, circularity, mean intensity and intensity std. dev. to identify the IP transition by searching for two clusters that minimize the within-class variance in the frames before the chromatin masses separate using the temporally constrained combinatorial clustering (TC3) method [17]. If the division time point that was identified during tracking corresponds to early anaphase, the software can automatically reposition the MA transition. Detecting early anaphase is accomplished by a heuristic that checks if the centroid distance of the chromatin masses of both daughter cells exceeds a user-defined threshold. As the classical method is originally applied to all trajectories, it may happen that the project still contains invalid trajectories. The second method is similar to the first method, but uses an additional auto-rejection of erroneous tracks. This is accomplished with a trainable LSTM network [46] that assesses the validity of each trajectory as a whole. The third method uses another LSTM network that was trained on sequences of CNN features that were obtained from the pretrained GoogLeNet to predict the state sequence for all-time points, as well as identifying which of the cell tracks are valid/invalid. The predicted synchronization time points are post-processed with a Hidden Markov Model (HMM) that only allows valid state transitions (*e.g.*, state sequences $1 \rightarrow 2 \rightarrow 3$ for a valid track or 0 for an invalid track) [47]. Transition probabilities are manually specified and based on the predicted states of the LSTM, and we use the Viterbi algorithm to identify the most likely hidden state sequence [48].

To inspect and optionally correct the automatic synchronization results, we provide a simple graphical user interface to manually identify the state transitions (Fig 1). This facilitates man-machine feedback, decreasing the size of the classifiers, and training time, if needed. It displays a set of cells, where two cells above one another are daughters and image snippets are preloaded to smoothly interact with the GUI. A manual annotation of two daughter cells can be accomplished with two clicks by identifying the last frame considered as interphase to mark the IP transition and the early anaphase frame to mark the MA transition. All intermediate frames are classified accordingly, and the annotations of one of the daughter cells are directly copied to the other daughter to have a consistent alignment. The GUI also allows rejecting entire trajectories, *e.g.*, if no mitotic event is present or due to erroneous tracking. The manually synchronized cells can additionally be used for retraining the LSTMs of the automatic

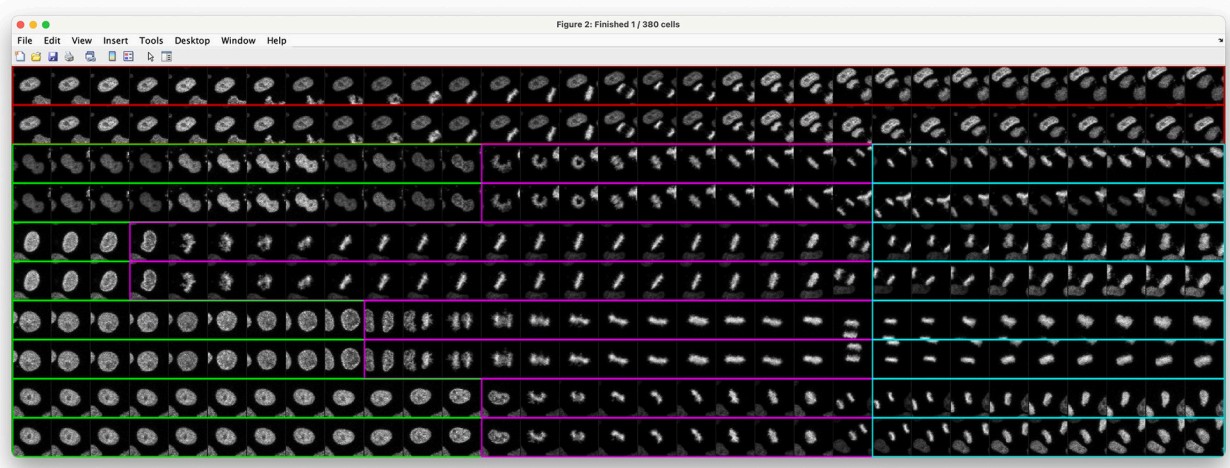

**Fig 1. Graphical user interface for trajectory synchronization.** Using an intuitive annotation scheme, users can verify and potentially correct the cell synchronization. Moreover, erroneous tracks that do not contain a mitotic event can be discarded. An initial synchronization can be automatically obtained using both classical and machine learning-based synchronization methods as detailed in the main text. Colors indicate interphase frames (green), pro-, prometa-, meta- and early anaphase frames (magenta), late ana- and telophase (cyan) and erroneous trajectories (red).

synchronization methods, and it is possible to specify separate models for different experimental conditions. We provide thorough validation of the different synchronization possibilities in S1–S5 Figs and in S4 Table. Once all cells are properly aligned, both qualitative and quantitative comparisons between different experiments can be performed, as detailed in the next sections.

## Data selection and visualization capabilities

An important aspect for the analysis of a particular subset of cells is data selection. This can be accomplished by using class-based selection procedures that allow to group the data according to imported metadata. For instance, it is possible to select experiments that were acquired with a particular microscope, a subset of treatments, a specific experiment or individual positions. It is also possible to use multiple properties in combination, to specify a feature range for selection, and to use the basic functionality of SciXMiner to add additional groupings derived from the individual or time series features [34]. Subsequent visualization, quantification and manual corrections are then automatically constrained to the selected cells.

In addition to the standard visualizations available in SciXMiner, we provide dedicated visualizations for the LiveCellMiner toolbox. Time series can be visualized as heatmaps, mean time series and combined line plots (Fig 2A–2C). In the heatmap visualization, each line represents a feature time series of a single cell, with feature values indicated by the color code. The identified synchronization time points are used to properly align the cell tracks below each other. Rather than displaying each cell separately, the mean ± std. dev. plots average the results of a particular position, experiment or microscope (Fig 2B). Averaging is performed on the aligned tracks to ensure that only corresponding mitotic stages are compared. In addition to presenting a single plot per selected group, it is also possible to combine all line plots including error bars in a single plot for better comparison (Fig 2C). Extracted single features can be visualized as box or violin plots (Fig 2D) and as histograms (Fig 2E).

All visualizations and subplots can be adjusted according to the selected grouping of the data. Aside from plotting all individual results in separate subplots, this allows combining

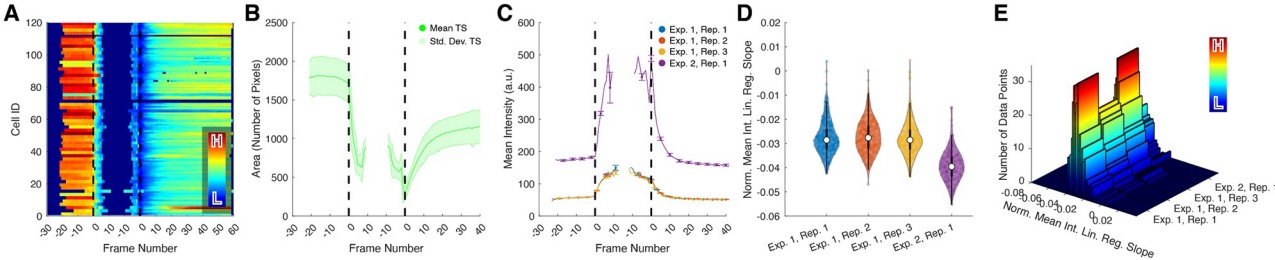

**Fig 2. Visualization options of LiveCellMiner.** The LiveCellMiner extension provides multiple ways of data visualization. (A) Temporally aligned heatmaps of feature time series (color encodes the feature value, *e.g.*, the area in numbers of pixels as in this example), (B) mean ± std. dev. curves summarizing all trajectories of a particular position or experiment in a separate subplot, (C) mean curves of multiple experiments with error bars plotted in a single axis for better comparability, (D) violin or box plots of single feature values and (E) histograms of individual data points grouped according to the current selection. See S12 Fig for more information about the time axis used for plotting temporal features.

related experiments, *e.g.*, visualizing the average response of a particular treatment across experiments, summarizing different repetitions of the same experiment or averaging responses across experiments. An example of three possible grouping scenarios is depicted in Fig 3.

Although it is good to have a variety of features that users can freely select from, not all features are necessarily relevant for characterizing a specific phenotype of interest. In addition to the targeted visualizations, LiveCellMiner thus also provides an easy way to obtain a ranked overview of all extracted features including statistical readouts like minimum, maximum, mean, standard deviation and median in table form as well as graphically using heat maps and combined line plots for all features. To identify which features are potentially suitable for characterizing phenotypic differences, we compute the n-fold change of each feature from the interphase average to the average of the first two prophase frames and the first two anaphase frames (window for averaging can be changed by the user). The generated report is exported in HTML format and can be conveniently displayed with any conventional internet browser (see S1 and S2 Files for a demonstration of automatically generated reports). Last but not least, LiveCellMiner can be used for statistical analysis of selected cells based on the single features

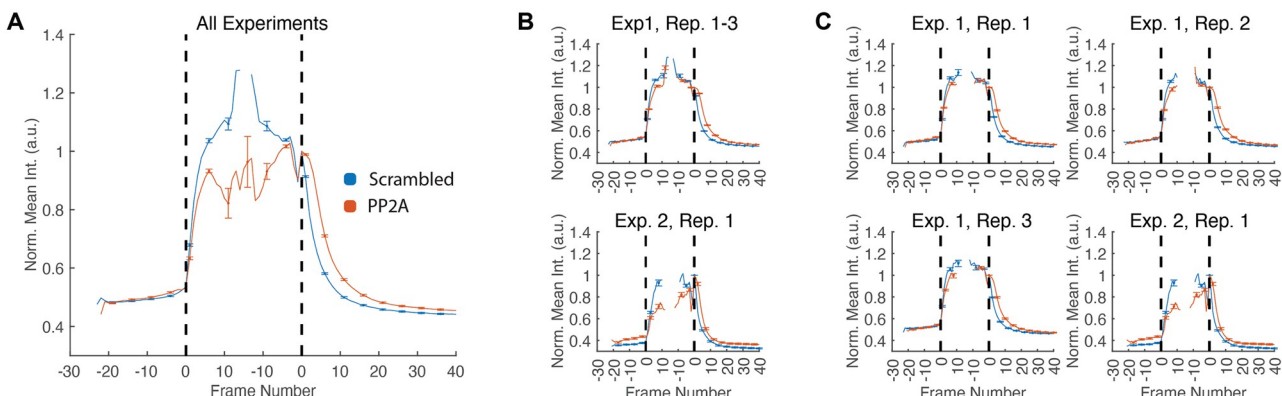

**Fig 3. Grouped data visualizations.** The three panels illustrate the different grouped visualization possibilities and were obtained using four experiments showing the normalized mean intensity for two oligos (Scrambled, PP2A). Exp. 1, Rep. 1–3 are three repetitions of the same experiment and Exp. 2, Rep. 1 is one separate experiment that was acquired using a different modality (confocal instead of a widefield microscope). The settings for combining the experiments (from left to right) are: (A) average time series of all experiments and repeats, (B) time series averaged over the repeats with separate plots per experiment and (C) individual plots for all experiments and repeats. Error bars indicate one standard deviation and the vertical bars represent the IP and the MA transitions.

or based on time series features of each cell. The methods for comparing single features across different groups comprise both parametric (two-sample t-test, ANOVA) and non-parametric tests (Wilcoxon, Kruskal-Wallis) as well as a two-way ANOVA (treatment × time) to be applied on selected time series. Results are exported as easily accessible spreadsheet files including the test results and the p-values of the performed tests.

## Results

### Cross-platform reproducibility of LiveCellMiner readouts

To test the capabilities of LiveCellMiner on the output of common light microscopy systems, we have reexamined the mitotic progression of cells after RNAi-mediated downregulation with negative and positive controls from previously published [37–39] and unpublished data sets (LSM710 confocal, see S1 Text). Our routine positive control is the RNAi-mediated down-regulation of three subunits of the heterotrimeric PP2A complex: PPP2CA (catalytic subunit alpha), PPP2R1A (scaffold subunit alpha) and PPP2R2A (a regulatory subunit B55 alpha). This PP2A complex is involved in the control of mitotic spindle assembly [49], and the spindle assembly checkpoint [50] promotes mitotic exit, disassembly of the spindle-pole associated microtubules in anaphase, resumption of nucleo-cytoplasmic transport, reclustering Golgi apparatus and chromatin decondensation ([51]; see [52, 53] for review).

When this specific PP2A complex is downregulated, various defects of mitotic progression are observed: prolongation of early mitosis (prophase, prometaphase and metaphase), partial late anaphase arrest and delayed reestablishment of nucleo-cytoplasmic transport [51]. In addition, PP2A downregulation leads to faulty chromatin decondensation and results in a partial telophase arrest [38]. The analysis of phenotypes using LiveCellMiner demonstrates that the biological effects of a given treatment can be quantitatively extracted by measuring a basic set of image features (Fig 4), without the need for extensive training of phenotypic classifiers. This is done in a reproducible manner across different live-cell microscopy platforms. In this case, area and intensity measures were used as proxy for chromatin decondensation (Fig 4A–4C), which is, as reported, delayed upon PP2A knockdown. In a similar way, deviations from control values in nuclear geometrical descriptors, *i.e.*, major and minor axes (Fig 4D), might indicate deformations and irregularities due to altered cytoskeleton or chromatin regulation [54]. LiveCellMiner automatically detects interphase to prophase and metaphase to anaphase transitions as well as the degree of rotation of the chromatin mass (Fig 4F and 4G). These read-outs allow detecting important errors in early mitotic progression. Delayed anaphase onset might arise from persistent chromosome misalignment and/or deficient spindle function as well as spindle misorientation, which often lead to chromosomal instability (CIN), a common feature in many pathologies including cancer ([55, 56], for review see [57, 58]).

However, HeLa cells do not move much and display relatively stable morphology with clearly distinguishable cytological changes, which facilitates segmentation and tracking in time-lapse imaging experiments. As this is not necessarily common to other cell types, we have challenged LiveCellMiner to compare the phenotype after PP2A downregulation in RPE cells, which move much more than HeLa and have more heterogeneous and changing nuclear morphologies. Applying the same segmentation and tracking algorithms like for HeLa, LiveCell-Miner performed reasonably good in segmentation, tracking and synchronization (S4 Fig, S4 Table). The results shown in (S9 Fig) are in line with what was shown in HeLa cells above, supporting the functions of PP2A across cell lines and the usability of LiveCellMiner with different cell types regardless of cell morphology or mobility. We noticed an increased number of falsely detected cell divisions that were successfully suppressed after training a synchronization classifier (S4 Fig, S4 Table). In future versions, we could potentially extend the tracking algorithm of

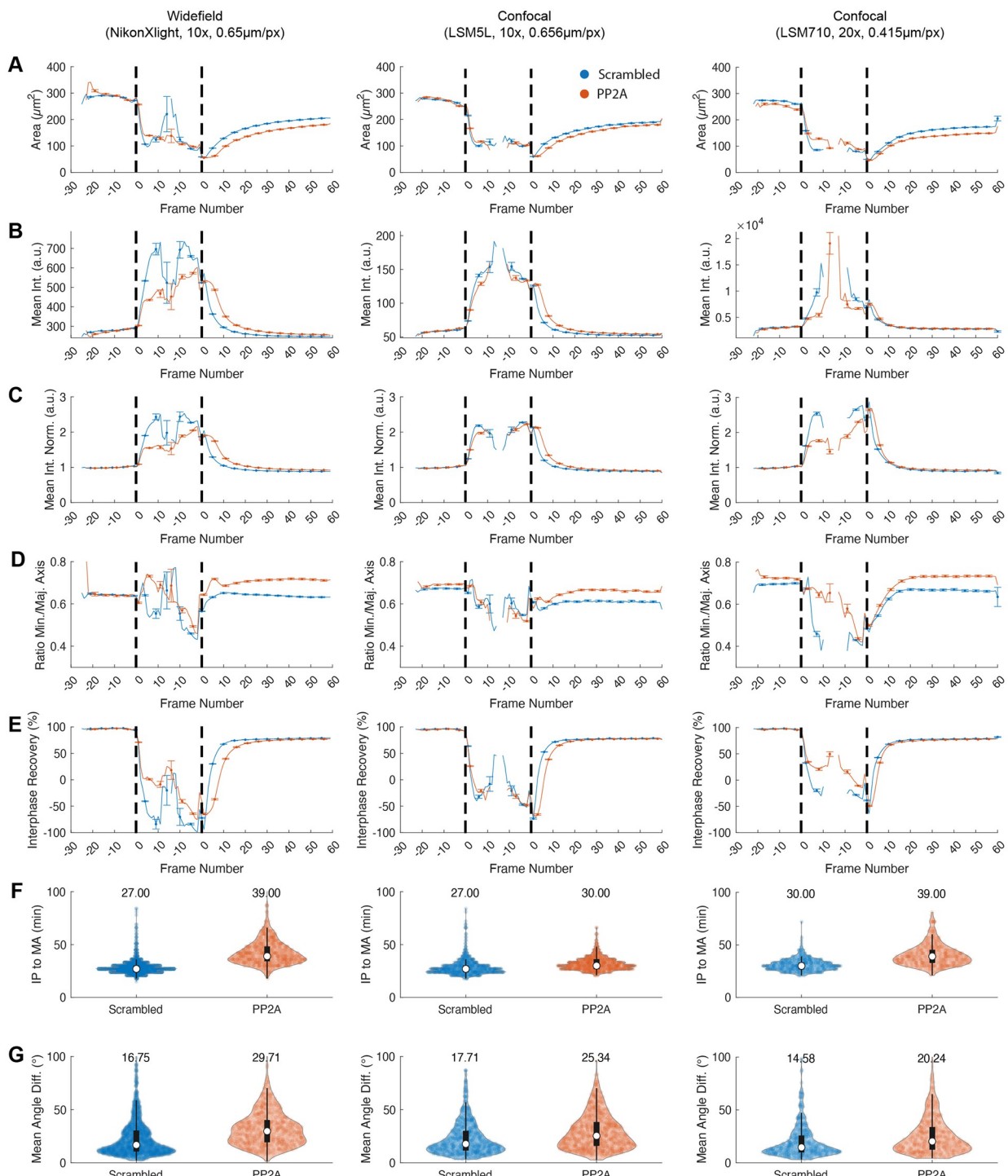

**Fig 4. Platform comparison.** Reproducibility study with different microscope systems. The columns show exemplary quantifications of the same experiment conducted on different microscopy platforms. We compare Scrambled (control, blue) vs. PP2A knockdown cells (orange). The time series features involve the chromatin area ($\mu m^2$) (A), the chromatin mean intensity (a.u.) (B), the normalized mean intensity (absolute intensity values divided by the interphase mean intensity of each cell, a.u.) (C), the minor axis vs. major axis ratio (D) and interphase-recovery feature as detailed in S2 Table (E). The violin plots show the duration between interphase-prophase and meta-anaphase transition in minutes (F) and the sum of the absolute angular changes in degrees (G). Widefield 10×: $N_{Scrambled} = 1262$, $N_{PP2A} = 1198$; Confocal 10×: $N_{Scrambled} = 2830$, $N_{PP2A} = 1008$; Confocal 20×: $N_{Scrambled} = 792$, $N_{PP2A} = 668$.

LiveCellMiner with a more complex division detection module to decrease the fraction of invalid tracks.

## Quantitative characterization of multiple mitotic phenotypes

To test LiveCellMiner for the quantitative study of a broad spectrum of mitotic phenotypes, we reanalyzed changes of chromatin and tubulin cytoskeleton appearance after LSD1 and RecQL4 RNAi-mediated downregulation in human cells using this tool.

We have previously described the Lysine Specific Demethylase, LSD1 (also known as KDM1A), as a crucial factor for reassembly of a functional nucleus at the end of mitosis [38]. Our recent work has shown that RecQ-like helicase 4 (RecQL4), whose mutations are causative of the Rothmund–Thomson syndrome, is important for stable chromosome alignment during mitosis [37]. These live-cell imaging experiments (S1 Text) were carried out in HeLa cells stably expressing H2B-mCherry, as chromatin marker, and eGFP-Tubulin for the spindle apparatus, which is the molecular machinery in charge of organizing and exerting the necessary forces to segregate chromatin.

Without the need of training experiment-specific phenotypic classifiers for the chromatin morphology or spindle apparatus, LiveCellMiner corroborates reduced chromatin decondensation rates, as area, mean intensity and interphase recovery in LSD1 downregulated cells (Fig 5A–5C and 5E). In turn, these image features that describe chromatin compaction state are unaffected by the RecQL4 downregulation (S8A–S8C Figs). The nuclear morphology can be also analyzed. In LSD1 and PP2A downregulated cells, after cell division, the nuclei become rounder (Fig 5D), as indicated by minor vs. major axis ratios. By contrast, after RecQL4 down-regulation, elongated nuclear shape is evident (Fig 5J). These anomalies might indicate an unbalance in the plethora of dynamic processes, factors and structures that reform the nuclear compartment during late mitosis (see [32, 59] for review). The molecular reasons and consequences of these, previously unnoticed, alterations in the nuclear morphology are not clear yet. However, abnormalities in the nuclear shape and architecture are widely observed in pathological conditions and ageing ([60] for review), which hint new paths for biomedical research regarding these protein targets. A manual confirmation of the observed phenotypes detected by LiveCellMiner is provided in S13 Fig and the same effects were observed in an independent experiment with RPE cells as shown in S9 Fig.

Previous work indicate that downregulation of PP2A, LSD1 or RecQL4 delays early mitotic progression. LiveCellMiner analysis shows the expected increase in the average time spent from prophase to anaphase onset after RNAi mediated downregulation of these targets (Fig 5G, 5H, 5M and 5N). However, a high degree of metaphase plate rotation is only observed in PP2A-downregulated cells, as the mean angle difference follows the axis of cell division (Figs 4G, 5I and 5O). This points to the different roles of PP2A and LSD1/RecQL4 in the control of the spindle apparatus during mitosis, consistent with current knowledge. The early mitotic delay after PP2A downregulation reflects broad defects in spindle function at the level of microtubule–kinetochore attachment [50] and bipolar spindle formation [49]. In turn, the abnormal fluctuations in the orientation of metaphase chromatin indicates faulty function of the cortical network and/or defective astral microtubules emanating from the spindle poles ([55, 56]; see [61] for review). In the case of LSD1 downregulation, where no metaphase chromatin rotation is observed, the early mitotic delay might arise from defects in chromatin methylation impairing correct chromosome segregation ([62]; see [63] for review), and might reflect unbalances in the expression of three major players of the mitotic control: PLK1[64], BUBR1 and MAD2[65], whose transcriptional regulation is influenced by LSD1. Likewise, the absence of rotation of the metaphase plate in RecQL4-downregulated cells suggests that the

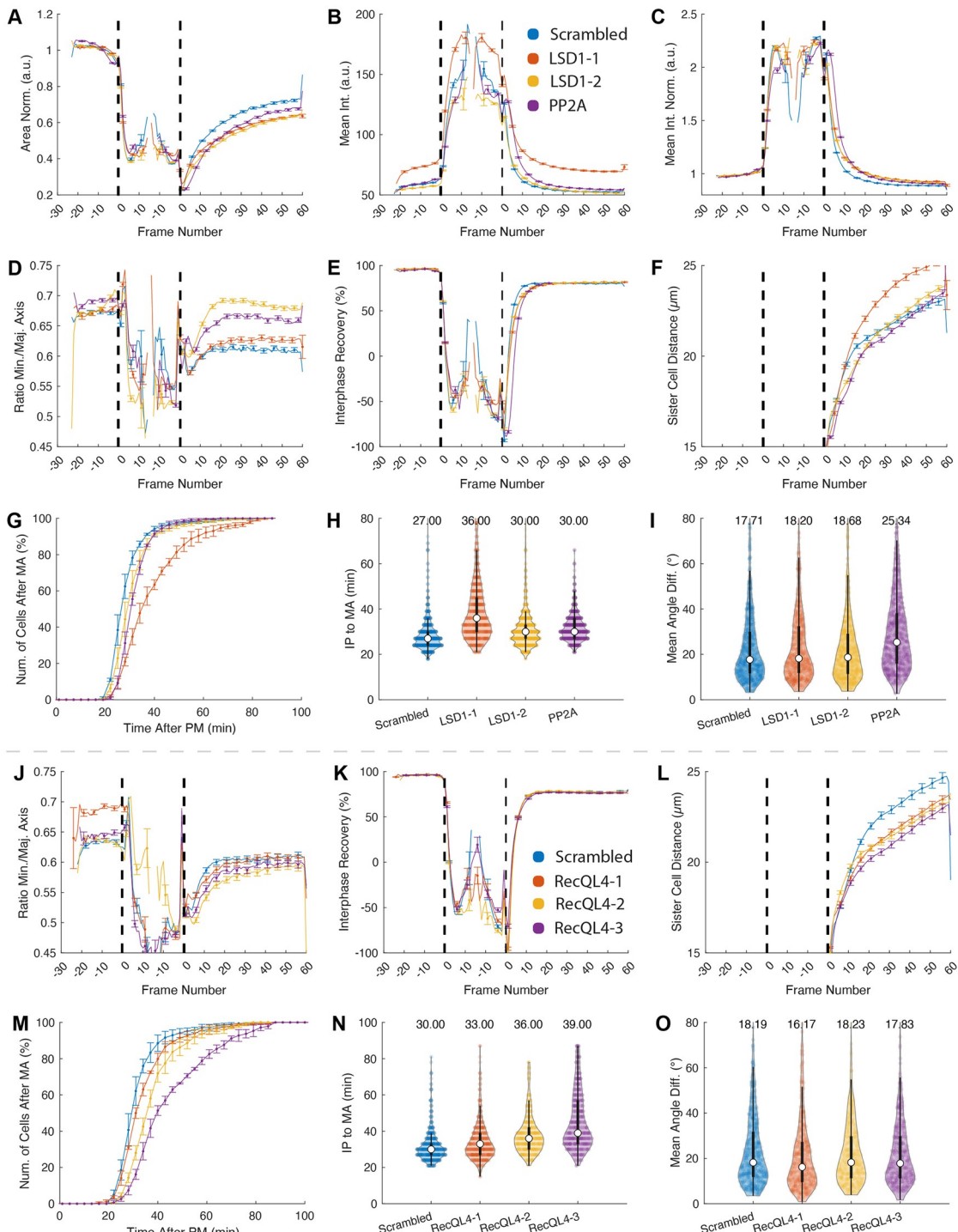

**Fig 5. Analysis of PP2A (Positive control), LSD1 and RecQL4 knockdowns.** Panels (A)-(I) show control (Scrambled) vs. LSD1–1, LSD1–2 and PP2A, whereas panels (J)-(O) show control (Scrambled) vs. RecQL4–1, RecQL4–2 and RecQL4–3 knockdown cells. The features involve the normalized area (A), the mean intensity (B), the normalized mean intensity (absolute intensity values divided by the interphase mean intensity of each cell, C), the minor axis vs. major axis ratio (D, J), the interphase recovery ratio (E, K), distance between sister chromatin masses (F, L) and cumulative histograms for the time in early mitotic progression until anaphase onset (G, M). The violin plots show the duration between interphase-prophase and meta-anaphase transition in minutes (H, N) and the mean angular difference in degrees (I, O). Images of panels (A)-(I) were acquired with a confocal microscope (LSM5L, 10×, 0.656$\mu m$/pixel). The plots combine extracted trajectories from three independent repeats with a total number of $N_{\text{Scrambled}} = 1262$, $N_{\text{LSD1-2}} = 970$, $N_{\text{LSD1-6}} = 1332$, $N_{\text{PP2A}} = 1198$ cells. Images of panels (J)-(O) below the dashed line were acquired

with a confocal microscope (LSM5L, 20*X*, 0.328*µm*/pixel). The plots combine extracted trajectories from three independent repeats with a total number of $N_{Scrambled} = 1094$, $N_{RecQL4-1} = 814$, $N_{RecQL4-3} = 842$, $N_{RecQL4-4} = 786$ cells. See S2 and S3 Tables for details on the depicted features.

delay in early mitotic progression is not due to malfunction of the astral microtubules and/or cortical network.

LiveCellMiner is able to measure the time-dependent distancing of daughter chromatin masses after anaphase onset. This readout is a proxy for chromosome and spindle dynamics during later mitotic stages. Consistent with known alterations in spindle dynamics during mitotic exit upon PP2A downregulation, delayed chromatin masses separation is observed early after anaphase onset (Fig 5F, S7D Fig). In turn, RecQL4 downregulated cells follow similar kinetics than control cells right after anaphase onset but decelerate the separation of the daughter nuclei later (Fig 5F and 5L). This might indicate that RecQL4 is involved in the complex regulation of cytoskeleton dynamics during mitotic exit and cytokinesis (see [66–68] for review), and opens a new research avenue.

Expressing fluorescent reporters in cells could perturb the phenomena under study. For example, overexpression of highly charged core histones can replace the surfactant effect that Ki-67 exerts in prometaphase in order to avoid excessive chromosome clustering [69]. Additionally, cell-to-cell variability can introduce artefacts affecting the dynamic range of the measure and preventing the detection of anomalies. To overcome these problems, LiveCellMiner permits single-cell temporal normalization of the extracted features. This allows extracting rates of change in comparable conditions (Fig 5B and 5C, Mean Int. vs. Mean Int. Norm, see also S8B and S8C Fig). Furthermore, to avoid or minimize the impact of cell-to-cell differences in expression of the reporters, LiveCellMiner includes a module to select cell sub-populations fulfilling certain criteria, *e.g.*, a limited intensity range of the H2B-mCherry chromatin marker in a given part of the cell cycle. For example, within the LSD1 data set, the oligo LSD1–1, but not the LSD1–2, increases considerably the amount of cells with extremely high H2B-mCherry signals (Fig 5A, Mean Int.). LiveCellMiner enables us to discard effects from the unequal reporter expression by confining the analysis, *i.e.*, to cells with low chromatin intensity values before entry into mitosis (S7 Fig).

The use of various fluorescent markers with different spectral properties is particularly useful for studying complex phenotypes. For mitosis-related events, studying the mitotic spindle by fluorescent labeling of different tubulin isoforms, *e.g.*, stable expression of eGFP-alpha-Tubulin, helps unraveling whether formation and/or function of the spindle apparatus is affected. Here, LiveCellMiner can extract spindle dynamics data by morphological dilation of the primary segmentation from the chromatin channel with a disk-shaped structuring element with a 15 pixel radius (see S3 Table for available features). PP2A but not RecQL4 or LSD1 RNAi-treated cells show a reliable increase of the spindle intensity staining during mitotic exit (S8D and S8G Fig) as well as a delayed disappearance of the astral spindle signal (detected as radial displacement of the intensity maximums in the GFP channel towards the polar regions) (S8E and S8F Fig). These measurements are consistent with previous findings which suggest that the PP2A-B55 holoenzyme regulates spindle-pole associated microtubules disassembly in late anaphase [51]. Furthermore, LSD1 but not PP2A (S8D and S8F Fig) or RecQL4 (S8G and S8I Fig) RNAi-treated cells show stronger spindle intensity during early mitotic progression. This previously unnoticed phenotype could hint to additional roles of LSD1 in spindle stability, finding that needs to be confirmed in the future.

## Reanalysis of mitotic phenotypes mediated by siRNA against VPS72, H2A. Z and the ATPase subunits from the chromatin remodeling complexes EP400, SRCAP and Ino80

We have recently shown that downregulation of VPS72, also referred to as YL-1, extends telophase in cells [39] by using live cell imaging combined with CecogAnalyzer 1.5.2. [10] analysis. VPS72 is part of the EP400 and the Snf2-related CBP-activator protein (SRCAP) chromatin remodeling complexes, where it works as a chaperon for H2A.Z. We also screened for telophase phenotypes after siRNA-mediated downregulation of EP400, SRCAP and H2A.Z, including as control the chromatin remodeling complex INO80, which does not contain VPS72. Here, we use the published data set to demonstrate the flexibility and suitability of LiveCellMiner for the analysis of high-content screening (HCS) data sets (S1 Text). Box-and-whisker or violin plots allow for comprehensive visualization of biologically relevant image features grouped by treatment, replicate number, etc. to enable direct comparison between dozens of experiments and allowing identification of samples with phenotypic deviations from a huge data set.

Reanalysis of the data set with LiveCellMiner revealed, for example, that downregulation of INO80, SRCAP, EP400 and H2A.Z consistently lengthens early mitotic progression compared to the scramble control, similar to the PP2A positive control (S6A Fig). In contrast, this is not observed upon VPS72 downregulation. These observations support previous findings where INO80 [70] and SRCAP [71] associate with the spindle apparatus and are required for proper mitotic progression. To our knowledge EP400 has not been linked to spindle function yet. However, our reanalysis raises the possibility that the EP400 chromatin remodeling complex is also involved in mitotic processes. Furthermore, the slight extension of early mitosis in H2A.Z depleted cells might indicate defective chromosome capture by the spindle apparatus due to chromosome centromeres alteration, where H2A.Z acts as organizer [72].

In a similar way, the detailed analysis of single features like chromatin area at different times (here, 21 and 42 min after anaphase onset) reveals that in PP2A- and VPS72-downregulated cells, nuclei at the end of mitosis are consistently smaller than in control cells (S6B and S6C Fig). In turn, PP2A siRNA-treated cells show additionally a delay in the rate of recovery of interphase average area (S6D and S6E Fig). This is consistent with the function of PP2A in the disassembly of spindle-pole associated microtubules and the reinitiation of nucleo-cytoplasmic transport in anaphase [51].

With this comprehensive display, intra- and inter-experimental variability can be analyzed (S6 Fig). For example, inter-experimental variability (for EXP1 and EXP2; see (S6D and S6E Fig) legend) for chromatin area and area recovery rates at 21 and 41 min are observed for the treatments with the PP2A-siRNA but not for scramble and the other siRNAs.

The concept of high-content screening implies that thousands of targets are tested in parallel. Often these setups require imaging for more than three days and slower frame frequency (*i.e.*, 5 to 8 min), which is needed to avoid phototoxicity and to provide enough time for the imaging loop through hundreds of positions. Longer times between frames might negatively impact segmentation and tracking performance resulting in inefficient phenotype recognition. Thus, we sought to test how LiveCellMiner performs with very large data sets of live-cell image sequences. For this, we took advantage of an original data set where they screened for genes involved in chromosome condensation at the beginning of mitosis in HeLa cells recorded every 8.5 min [33]. Constrained by our computing capacity, we reanalyzed only 76 positions of this HCS data set corresponding to solid-phase transfection of siRNA against scramble controls and two well-known mitotic regulators: CDC20 and Aurora Kinase A (AurKA). LiveCellMiner performed reasonably well in segmentation and tracking (S5 and S12 Figs, S4 Table),

accomplishing full analysis of each position in about 30 min with a single workstation. Processing time could be considerably reduced by data-parallel computing, *e.g.*, by distributing the positions of a screen to separate nodes of a computing cluster. In line with available knowledge, LiveCellMiner detects a delay in the early mitotic progression for siRNA mediated downregulation of CDC20 [73] and AurKA [74] (S10C Fig). Our analysis also reflects the known role of CDC20 during mitosis exit [10] as a delay in chromatin expansion after anaphase onset in CDC20 downregulated cells (S10A, S10B, S10D and S10E Fig). These results, along with those in S9 Fig, demonstrate the ability of LiveCellMiner to successfully and rapidly analyze data, across a range of cell types and frame frequencies, including those from HCS.

In the future, we will validate the hypotheses formulated here and investigate the molecular mechanisms behind the newly described early mitotic phenotypes. In essence, the reevaluation of the live cell imaging data sets confirmed previous findings regarding late mitotic phenotypes. These examples illustrate the power of LiveCellMiner to screen complex phenotypes in a quantitative manner using simple to complex live-cell imaging experiments. Our examples focus on mitotic chromatin, but with modest modifications, LiveCellMiner would also be applicable to other fluoresently labeled subcellular structures.

## Availability and future directions

The LiveCellMiner extension package can be obtained from the following repository https://github.com/stegmaierj/LiveCellMiner. We provide detailed installation instructions and an overview of all LiveCellMiner-specific functions on the landing page of the repository. As potential improvements will be made available via the repository, a snapshot of the latest version upon the submission time point can be obtained here https://zenodo.org/badge/latestdoi/269630703.

While LiveCellMiner provides already all tools to perform in-depth analyses of high-content screens, there are a few points we will address in future versions of the software. In the current implementation, LiveCellMiner searches for mitotic events and uses the anaphase onset as a reference for extracting the remaining frames for the analysis. In future versions, we will also add the possibility to analyze non-mitotic tracks and adapt the synchronization tools to other scenarios as well. Although LiveCellMiner's synchronization tools were developed for recognizing mitotic morphologies of mammalian cells, the recognition of mitotic transitions in other organisms is possible. For instance, the analysis of mitotic progression of yeast could be easily implemented in the code, given that the shape of the chromatin mass is very particular at anaphase and could be supported by measurements of the position of the nucleus with respect to the yeast walls. The current detection and segmentation methods sequentially process individual images one at a time. To speedup processing for larger screens these steps could potentially be parallelized to fully exploit multicore CPUs and GPUs as available in the respective system. As a temporary workaround, one can run multiple instances of LiveCellMiner and thereby distribute the processing of independent projects, *e.g.*, on different cluster nodes. The deep learning-based segmentation relies currently on the external pretrained software tool Cellpose [16]. The advanced segmentation methods could be implemented directly in MATLAB to streamline the processing with as little additional dependencies as possible. Finally, the trajectory synchronization module of LiveCellMiner currently involves a few semi-automatic steps that can become time-consuming for very large and highly diverse screens. A future avenue of research will thus be improving the reliability of unsupervised approaches like [17] to ultimately analyze high-content screens in a fully-automatic fashion.

## Supporting information

**S1 Text. Experimental details.**
(PDF)

**S1 Fig. Validation of the synchronization performance for a confocal data set.** The manually annotated data comprise three experiments with two positions each (20× objective, physical spacing 0.415$\mu m$, 3 minute time intervals and $N$ = 832 trajectories in total). Columns show the results of the three different synchronization methods (Classical, Classical+Auto Rejection and LSTM+HMM+Auto Rejection) as described in the main text. The first two rows show histograms of the frame offset of the automatically identified synchronization time points with respect to a manually annotated ground truth (IP: interphase to prophase transition and MA: metaphase to anaphase transition). For instance, a value of 1 indicates that the synchronization time point is set one frame too late and 0 indicates a perfect match. The last row quantifies precision, recall, f-score and accuracy of the auto rejection module that is intended to discard erroneous tracks. Validation was performed using a 6-fold cross validation by retraining the LSTM-based classifier on each split and testing the classifier on each of the remaining test data.
(TIF)

**S2 Fig. Validation of the synchronization performance for a widefield data set.** The manually annotated data comprise three experiments with two positions each (10× objective, physical spacing 0.65$\mu m$, 3 minute time intervals and $N$ = 832 trajectories in total). Columns show the results of the three different synchronization methods (Classical, Classical+Auto Rejection and LSTM+HMM+Auto Rejection) as described in the main text. The first two rows show histograms of the frame offset of the automatically identified synchronization time points with respect to a manually annotated ground truth (IP: interphase to prophase transition and MA: metaphase to anaphase transition). For instance, a value of 1 indicates that the synchronization time point is set one frame too late and 0 indicates a perfect match. The last row quantifies precision, recall, f-score and accuracy of the auto rejection module that is intended to discard erroneous tracks. Validation was performed using a 6-fold cross validation by retraining the LSTM-based classifier on each split and testing the classifier on each of the remaining test data.
(TIF)

**S3 Fig. Validation of the synchronization performance for the LSD1 data set.** The manually annotated data comprise three experiments with 16 positions each (10× objective, physical spacing 0.656$\mu m$, 3 minute time intervals and $N$ = 5878 trajectories in total). Columns show the results of the three different synchronization methods (Classical, Classical+Auto Rejection and LSTM+HMM+Auto Rejection) as described in the main text. The first two rows show histograms of the frame offset of the automatically identified synchronization time points with respect to a manually annotated ground truth (IP: interphase to prophase transition and MA: metaphase to anaphase transition). For instance, a value of 1 indicates that the synchronization time point is set one frame too late and 0 indicates a perfect match. The last row quantifies precision, recall, f-score and accuracy of the auto rejection module that is intended to discard erroneous tracks. Validation was performed by training on 1328 trajectories that were evenly distributed among all positions and by applying it to a remaining set of 2990 trajectories.
(TIF)

**S4 Fig. Validation of the synchronization performance for the RPE data set.** The manually annotated data comprise one experiment with 6 positions (Nikon microscope, 20x objective,

physical spacing 0.33$\mu m$, 3 minute time intervals and $N = 388$ trajectories in total). Columns show the results of the three different synchronization methods (Classical, Classical+Auto Rejection and LSTM+HMM+Auto Rejection) as described in the main text. The first two rows show histograms of the frame offset of the automatically identified synchronization time points with respect to a manually annotated ground truth (IP: interphase to prophase transition and MA: metaphase to anaphase transition). For instance, a value of 1 indicates that the synchronization time point is set one frame too late and 0 indicates a perfect match. The last row quantifies precision, recall, f-score and accuracy of the auto rejection module that is intended to discard erroneous tracks. Validation was performed using a 6-fold cross validation by retraining the LSTM-based classifier on each split and testing the classifier on each of the remaining test data.
(TIF)

**S5 Fig. Validation of the synchronization performance for the public data set by Hériché et al.[33].** The manually annotated data comprise twelve experiments with 4 positions each (automated epifluorescence microscope, 20× objective, physical spacing 0.32$\mu m$, 8.5 minute time intervals and $N = 2200$ trajectories in total). Columns show the results of the three different synchronization methods (Classical, Classical+Auto Rejection and LSTM+HMM+Auto Rejection) as described in the main text. The first two rows show histograms of the frame offset of the automatically identified synchronization time points with respect to a manually annotated ground truth (IP: interphase to prophase transition and MA: metaphase to anaphase transition). For instance, a value of 1 indicates that the synchronization time point is set one frame too late and 0 indicates a perfect match. The last row quantifies precision, recall, f-score and accuracy of the auto rejection module that is intended to discard erroneous tracks. Validation was performed using a 6-fold cross validation by retraining the LSTM-based classifier on each split and testing the classifier on each of the remaining test data.
(TIF)

**S6 Fig. Exemplary readouts from a high-content screen.** The different panels show violin plots of the IP to MA duration in minutes (A), the area at 21 and 42 minutes after anaphase onset (B, C), the area recovery compared to the level at interphase in % at 21 and 42 minutes after the anaphase onset (D, E) as well as a combined recovery measure comprised of area, minor axis length, mean intensity and intensity standard deviation at 60 minutes after anaphase onset (F). See S2 Table for a more detailed description of the individual features. Numbers above each violin are the respective median values.
(TIF)

**S7 Fig. Analysis of LSD1 with constrained intensity range (40-46).** Images were acquired with a confocal microscope (LSM5L, 10×, 0.656$\mu m$/pixel). We compare control (Scrambled), LSD1-1, LSD1-2 and PP2A knockdown cells. The basic features involve the normalized area (A), the mean intensity (B), the normalized mean intensity (C) and the sister cell displacement (D). The violin plots show the duration between interphase-prophase and metaphase-anaphase transition in minutes (E) and the mean orientation angle difference in degrees (F). The selection was constrained to cells exhibiting an interphase mean intensity in the range of 40−46, which yielded a set of $N_{Scrambled} = 84$, $N_{LSD1-1} = 18$, $N_{LSD1-2} = 190$, $N_{PP2A} = 106$ cells.
(TIF)

**S8 Fig. Analysis of LSD1 and RecQL4 knockdowns (Additional features).** Panels (A)-(F) show control (Scrambled) vs. RecQL4-1, RecQL4-2 and RecQL4-3, whereas panels (G)—(I) show control (Scrambled) vs. LSD1-1, LSD1-2 and PP2A knockdown cells. The features involve the normalized area (A), the mean intensity (B), the normalized mean intensity

(absolute intensity values divided by the interphase mean intensity of each cell, C). Panels (D-I) exemplify features that were extracted from the second fluorescence channel and include the normalized mean intensity (D,G), the distance of the intensity maximum to the segmentation centroid (E,H) and the average mean intensity between the IP and MA transitions (F,I). Images of panels (A)-(F) above the dashed line were acquired with a confocal microscope (LSM5L, 20X, 0.656$\mu m$/pixel). The plots combine extracted trajectories from three independent repeats with a total number of $N_{\text{scrambled}} = 1094$, $N_{\text{RecQL4-1}} = 814$, $N_{\text{RecQL4-3}} = 842$, $N_{\text{RecQL4-4}} = 786$ cells. Images of panels (G)-(I) below the dashed line were acquired with a confocal microscope (LSM5L, 10×, 0.656$\mu m$/pixel). The plots combine extracted trajectories from three independent repeats with a total number of $N_{\text{Scrambled}} = 1262$, $N_{\text{LSD1-2}} = 970$, $N_{\text{LSD1-6}} = 1332$, $N_{\text{PP2A}} = 1198$ cells. See S2 and S3 Tables for details on the depicted features.
(TIF)

**S9 Fig. Analysis of PP2A knockdowns in RPE cells.** Images were acquired 48 h post transfection with with 20nM siRNA by the widefield module of a Ti2 Eclipse (Nikon) equipped with a LED light engine SpectraX (Lumecor) and GFP/mCherry filter sets, a Plan-Apochromat 20x NA 0.75 and scaling 0.33μm/pixel. We compare the quantitation of features as in Fig 5 for control (Scrambled) and PP2A knockdown. The plots combine extracted trajectories from $N_{\text{Scrambled}} = 104$ and $N_{\text{PP2A}} = 20$ cells. See S2 and S3 Tables for details on the depicted features.
(TIF)

**S10 Fig. Analysis of CDC20 and Aurora Kinase A knockdowns in HeLa cells from the HCS data set published by Hériché *et al.*[33].** This data set is publicly available at Image Data Resource (IDR) (https://idr.openmicroscopy.org/webclient/?show=screen-102). There, HeLa cells stably expressing HIST1H2BJ-mCherry and LMNA-eGFP were cultured in siRNA-coated 96-well plates. The images were acquired with an Olympus IX-81 automated epifluorescence microscope with a 20× objective, physical spacing 0.32μm and a time interval of 8.5 min for 44 h. Four independent replicates were acquired for each siRNA treatment. The plots show pooled measures from 48 scrambled-, 24 siCDC20- and 4 AurKA-1- siRNA treated positions ($N_{\text{Scrambled}} = 668$, $N_{\text{CDC20}} = 32$, $N_{\text{AurKA-1}} = 86$). See S2 and S3 Tables for details on the depicted features.
(TIF)

**S11 Fig. Illustration of the LoG-based nucleus detection and cellpose segmentation.** Cellpose provides highly accurate nuclei segmentation for most of the cells. However, in some rare cases (*e.g.*, in late anaphase), cells tend to flicker and remain undetected for one or more frames. To prevent interrupted tracks for such misdetections, LiveCellMiner provides a fallback option on classical image analysis methods and uses a LoG-based nucleus detection coupled with a classical binary threshold and watershed-based segmentation as detailed in the main text. The depicted examples qualitatively demonstrate the accurate segmentation performance of Cellpose and a few examples where detections were missed that are successfully identified by the classical LoG-based detection method. We found that using both approaches in combination resulted in complementary results and effectively in more complete tracks as quantitatively demonstrated in S1 Table. The average diameter of all cells and across all time points in this example is 40.79 pixels and the Cellpose diameter parameter was set to the default value of 30 pixels, to allow segmenting smaller objects like cells in meta- and anaphase as well.
(TIF)

**S12 Fig. Examples of temporally aligned cells from different experiments and time line explanations.** (A) Each row shows a single cell of different experiments (see legend in the

figure for imaging details). Images were scaled to a consistent size, temporally aligned and contrast was adapted for better visibility. (B) In all temporal plots of the manuscript, we group the visualization into three different phases, (1) interphase, (2) pro-, prometa-, meta- and early anaphase and (3) late ana- and telophase. The dashed lines indicate the interphase to prophase transition (IP) and the metaphase to anaphase transition/anaphase onset (MA). For better visibility and to be able to visualize cells with different IP to MA duration, the frames between IP and MA are evenly distributed to the left and right. Time stamps are in minutes and display the relative timing with respect to the transition time points.
(TIF)

**S13 Fig. Manual validation of unexpected phenotypes in LSD1 and PP2A depleted cells.** To confirm the unexpected nuclear morphology phenotypes in LSD1 and PP2A in late mitosis detected by LiveCellMiner, we manually measured the major/minor axes (A, C), circularity (B) and the mean intensity of cells (D) from the original LSD1 data set records 45h post transfection. Using Fiji [75], 10 random daughter cells 30 min after anaphase onset per condition per experiment (three independent experiments indicated in colored data points) were manually segmented with the Wand tool in legacy mode, and the values were extracted using the ROI manager. The data were organized in excel tables and plots (box and whisker min to max) and statistical analysis were done using GraphPath (D'Agostino-Pearson normality test and parametric one-way ANOVA).
(TIF)

**S1 Table. Quantification of the detection, segmentation and tracking performance.** Measurements were performed on two manually annotated time series of the `Fluo-N2DL-HeLa` data set that is part of the Cell Tracking Challenge [76] and was most similar to our application scenario (acquisition was performed with an Olympus IX81 microscope using a 10×/0.4 objective lens, a physical spacing of 0.645×0.645$\mu m$ and 30 minute time intervals). The used metrics are DET (detection score), SEG (segmentation score), TRA (tracking score), $OP_{CTB}$ (average of SEG and TRA) and $OP_{CSB}$ (average of DET and SEG). All metrics have a value range between 0 and 1 with 1 being the optimum (see [76] for details on the measures). While Cellpose [16] yields better segmentation scores, it misses cells occasionally. On the other hand, our classical approach based on multi-scale Laplacian-of-Gaussian blob detection is able to find most cells with a slightly worse segmentation accuracy. Combining both approaches, *i.e.*, complementing the Cellpose segmentation with additional cells that were only found by the classical method, yielded consistently the best results (highlighted in bold face letters). The experiments described in the main paper were consistently acquired with 3 minute time intervals, which further improves the tracking performance. Please note that neither parameter tuning of the classical method nor any retraining of Cellpose was performed. Thus, the top-scoring methods listed on the Cell Tracking Challenge leader board are still slightly higher but we expect our method to generalize better to unseen data sets.
(XLSX)

**S2 Table. Description of the relevant time series (TS) and single features (SF) used for characterizing cell behavior based on the chromatin channel.** We used MATLAB's `regionprops` function for basic morphological features [77]. Moreover, a detailed description of the Haralick features can be found in [43] and we used the implementation described in [78].
(XLSX)

**S3 Table. Description of the relevant time series (TS) and single features (SF) that can be extracted from the second channel (optionally, if available).** Note that all features mentioned in S2 Table can also be extracted for the secondary channel and are not listed here.

Moreover, each of the listed features can be computed on a grown/shrunk/toroidal region obtained via morphological dilation/erosion of the chromatin mass segmentation with a user-defined radius and structuring element.
(XLSX)

**S4 Table. Quantification of the synchronization performance.** Measurements were performed on a confocal data set (LSM710, 20× objective, physical spacing 0.415$\mu m$, 3 minute time intervals and $N = 1606$ trajectories in total), a widefield data set (NikonXLight, 10× objective, physical spacing 0.65$\mu m$, 3 minute time intervals and $N = 5248$ trajectories in total), a confocal data set (LSM5L, 10× objective, physical spacing 0.656$\mu m$, 3 minute time intervals and $N = 2990$ trajectories in total), the RPE data set (Nikon microscope, 20x objective, physical spacing 0.33$\mu m$, 3 minute time intervals and $N = 388$ trajectories in total) and the public data set by Hériché *et al.* [33] (automated epifluorescence microscope, 20× objective, physical spacing 0.32$\mu m$, 8.5 minute time intervals and $N = 2200$ trajectories in total). True positives (TP), true negatives (TN), false positives (FP) and false negative (FN) are summed over all folds of the 6-fold cross-validation. Precision, recall, accuracy and F-Score were computed individually on each fold and the displayed values are the mean ± SD. The last two columns contain the mean ± SD values of the synchronization time point offsets for the interphase prophase transition (IP) and the metaphase to anaphase transition (MA) measured in frames (a deviation of 1 frame corresponds to 3 minutes for both data sets). Bold-face letters indicate the best values for each measure and data set.
(XLSX)

**S5 Table. Statistical analysis.** Statistical test results for the violin plots of Figs 4 and 5 and S6–S8 Figs summarized in a supplementary spreadsheet file.
(XLSX)

**S6 Table. Statistical analysis.** Statistical test results for the time series plots of Figs 4 and 5 and S6–S8 Figs summarized in a supplementary spreadsheet file.
(XLSX)

**S7 Table. LiveCellMiner dependencies.** LiveCellMiner uses the previously published tools XPIWIT [36], Cellpose [16] and SciXMiner [34] for object detection, segmentation, project organization and GUI development, respectively. This table lists the third party dependencies of LiveCellMiner.
(XLSX)

**S1 File. Generated report for the LSD1 data set.** All existing single features and time series features are contained and accessible from an HTML-based overview file. Extract the archive to a folder of your choice and open the HTML file in the root directory using any web browser.
(ZIP)

**S2 File. Generated report for the RecQL4 data set.** All existing single features and time series features are contained and accessible from an HTML-based overview file. Extract the archive to a folder of your choice and open the HTML file in the root directory using any web browser.
(ZIP)

## Acknowledgments

We thank Wolfram Antonin for valuable comments and proofreading of the manuscript.

## Author Contributions

**Conceptualization:** Daniel Moreno-Andrés, Johannes Stegmaier.

**Data curation:** Daniel Moreno-Andrés, Anja Scheufen, Johannes Stegmaier.

**Formal analysis:** Daniel Moreno-Andrés, Johannes Stegmaier.

**Funding acquisition:** Daniel Moreno-Andrés, Johannes Stegmaier.

**Investigation:** Daniel Moreno-Andrés, Johannes Stegmaier.

**Methodology:** Daniel Moreno-Andrés, Anuk Bhattacharyya, Johannes Stegmaier.

**Project administration:** Daniel Moreno-Andrés, Johannes Stegmaier.

**Resources:** Daniel Moreno-Andrés, Anja Scheufen, Johannes Stegmaier.

**Software:** Anuk Bhattacharyya, Johannes Stegmaier.

**Supervision:** Daniel Moreno-Andrés, Johannes Stegmaier.

**Validation:** Daniel Moreno-Andrés, Johannes Stegmaier.

**Visualization:** Daniel Moreno-Andrés, Johannes Stegmaier.

**Writing – original draft:** Daniel Moreno-Andrés, Johannes Stegmaier.

**Writing – review & editing:** Anuk Bhattacharyya, Anja Scheufen.

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
