## [Decision Letter · Decision Letter 0]

28 Apr 2022

PONE-D-22-08004LiveCellMiner: A New Tool to Analyze Mitotic ProgressionPLOS ONE

Dear Dr. Stegmaier,

Thank you for submitting your manuscript to PLOS ONE. After careful consideration, we feel that it has merit but does not fully meet PLOS ONE’s publication criteria as it currently stands. Therefore, we invite you to submit a revised version of the manuscript that addresses the points raised during the review process.

We look forward to receiving your revised manuscript.

Kind regards,

Yanchang Wang

Academic Editor

PLOS ONE

Journal Requirements:

Additional Editor Comments:

Your manuscript was reviewed by one reviewer and the response from this reviewer is very positive. This reviewer expressed only minor concerns. Specifically, I agree with the reviewer that images of nuclear morphology of cells with downregulated LSD1 and PP2A need to be shown. Other minor issues are also expected to be addressed in the revised version.

Reviewers' comments:

Reviewer's Responses to Questions

**Comments to the Author**

1. Is the manuscript technically sound, and do the data support the conclusions?

Reviewer #1: Yes

2. Has the statistical analysis been performed appropriately and rigorously? 

Reviewer #1: Yes

3. Have the authors made all data underlying the findings in their manuscript fully available?

Reviewer #1: Yes

4. Is the manuscript presented in an intelligible fashion and written in standard English?

Reviewer #1: Yes

5. Review Comments to the Author

Reviewer #1: In this manuscript, Moreno-Andrés et al. describe an open-source, automated software tool designed to quantify cell-cycle progression using live-cell imaging data called LiveCellMiner. The main goal of this manuscript is to show that LiveCellMiner can be used to accurately measure the cell-cycle using live-cell imaging data across various platforms and conditions. Furthermore, the authors demonstrate that LiveCellMiner reproduces previously characterized data in cells depleted of various cell-cycle proteins including PP2A, LSD1, and RecQL4. Lastly, the authors adapt their software to analyze data generated from high-content screens.

This manuscript generally demonstrates the broad capabilities of this software to accurately analyze live-cell data. The demonstrated reproducibility across various microscopy platforms will increase the adoption by the scientific community. The software builds on many pre-existing methods. While the novelty of this software is lacking in this regard, the utility will ultimately be determined by the users. With the enhancements of this software to make it more user-friendly, I find it to be of great value to the scientific community.

Altogether, I find that LiveCellMiner will be a very useful tool for analyzing cell-cycle progression using live cell imaging data.

Specific comments:

1. The authors find that LiveCellMiner detected nuclear morphology phenotypes in LSD1 and PP2A depleted cells (line 331). This unexpected discovery needs to be validated to prove that the software accurately measures this phenotype. Because of this, the ability to use LiveCellMiner to uncover new phenotypes could be strengthened by showing that the abnormal nuclear shape detected by LiveCellMiner is observed in LSD1 and PP2A depleted cells.

2. I found the visualization options to be unclear with respect to the X-axis. Specifically, Fig.2 A-C, and similar figures. The work could benefit from a clear description of the X-axis and the vertical dotted lines in the legend. At least one description with corresponding images would be useful in the main text. (e.g. Fig.S12 paired with a graph).

3. The authors need to clearly state the types of cell lines that are used in the analysis. It is not until the comparison of Hela to RPE cells that it is implied that the presented results are mammalian cells. Is this software only limited to mammalian cells or can this be used for other species commonly used for cell-cycle research such as S. cerevisiae or S. pombe? Addition of this information would be useful to the scientific community.

4. What are the limitations in the data input in LiveCellMiner? The author’s comment: Line 87 “LiveCellMiner expects time series of 2D images of cells with chromatin-labeled nuclei and optionally additional markers in other channels.” Additional information about what the constraints on the input data would benefit those deciding to use the software.

5. Fig 5 title should also include PP2A.

6. PLOS authors have the option to publish the peer review history of their article (what does this mean?). If published, this will include your full peer review and any attached files.

Reviewer #1: No

---

## [Author Response · Author response to Decision Letter 0]

1 Jun 2022

Please see PDF "ResponseToReviewers.pdf" submitted along with the attached files.

---

## [Editor Report · Decision Letter 1]

20 Jun 2022

LiveCellMiner: A New Tool to Analyze Mitotic Progression

PONE-D-22-08004R1

Dear Dr. Stegmaier,

We’re pleased to inform you that your manuscript has been judged scientifically suitable for publication and will be formally accepted for publication once it meets all outstanding technical requirements.

Kind regards,

Yanchang Wang

Academic Editor

PLOS ONE
---

## [Editor Report · Acceptance letter]

27 Jun 2022

PONE-D-22-08004R1 

LiveCellMiner: A New Tool to Analyze Mitotic Progression 

Dear Dr. Stegmaier:

I'm pleased to inform you that your manuscript has been deemed suitable for publication in PLOS ONE. Congratulations! Your manuscript is now with our production department. 

Kind regards, 

on behalf of

Dr. Yanchang Wang 

Academic Editor

PLOS ONE